# A Regulatory Game Analysis of Smart Aging Platforms Considering Privacy Protection

**DOI:** 10.3390/ijerph19095778

**Published:** 2022-05-09

**Authors:** Tengfei Shi, Hanjie Xiao, Fengxia Han, Lan Chen, Jianwei Shi

**Affiliations:** 1Faculty of Management and Economics, Kunming University of Science and Technology, Kunming 650504, China; tengfeikmust@163.com (T.S.); chenlankust@163.com (L.C.); shijianweiiiii@163.com (J.S.); 2School of Economics and Management, Huzhou University, Huzhou 313000, China; kmustxiao2014@126.com; 3Industrial Development Research Institute, Kunming University of Science and Technology, Kunming 650031, China

**Keywords:** smart aging platforms, supervisory, secret protection, privacy disclosure, evolutionary game

## Abstract

Privacy and information protection are important issues in the era of big data. At present, China’s elderly care industry is gradually adopting the supply model of smart elderly care to alleviate the contradiction between supply and demand. However, the low level of regulation of smart aging platforms may lead to a low level of privacy protection on the platforms. Therefore, in this paper, based on the evolutionary game and Lyapunov theory, we discuss the willingness of elderly people to participate in regulation, the privacy protection status of platform service providers, and the degree of government regulation, as well as the key factors affecting the equilibrium of the three-party game system, and conduct simulation analysis and game system optimization using MATLAB. The simulation results show that $A_{1}(0,0,1)$ and $A_{5}(0,0,0)$ can be transformed to $A_{8}(1,1,0)$ by adjusting the parameters, i.e., the optimal ESS is participation, high-quality protection, and low investment supervision; the service income of the elderly, the loss of privacy leakage, the investment cost of service providers, and the amount of government rewards and punishments are the key factors affecting the tripartite game system. By analyzing the impact of factors, such as benefits and costs, on privacy protection and the regulation of smart senior care platforms, the level of privacy protection of smart senior care platforms can be improved and the process of the comprehensive regulation of domestic senior care services can be promoted.

## 1. Introduction

With our aging society becoming a more serious problem, the number of elderly care institutions is gradually increasing; however, the majority of them have a low level of information technology, and the elderly services and facilities are relatively simple and cannot meet the diverse medical and care needs of the elderly [1]. The Opinions of the General Office of the State Council on Promoting the Development of Senior Care Services states that it will implement the “Internet + Senior Care” initiative, continue to promote the development of the intelligent health and senior care industry, and promote the in-depth application of new-generation information technologies, such as artificial intelligence, the Internet of Things, cloud computing, big data, and other intelligent hardware in the field of senior care services [2]. The current elderly care service has met the needs of elderly families for home care services by reducing costs through online elderly care services; improving the efficiency of consultations and service quality for the elderly with the help of terminal devices, such as smartphones and tablets; and by using various smart senior care platforms, such as Love Family Elderly, Family Circle and Elderly Care Manager. However, when improving the quality of elderly care services, there are also more privacy and security issues, such as elderly-related apps infringing on the private information of users’ login usernames, passwords, and using medical and health data in other ways, and the existence of the app’s functions to obtain user privacy permissions for non-essential use, increasing the risk of privacy leaks for elderly users and their family members [3]. The 2021 Cost of Data Breach Report, published by IBM Security and the Ponemon Institute, shows that the average total cost of a data breach increased by nearly 10% year-on-year, the largest single-year cost increase in nearly seven years; the cost of a data breach increased from $3.86 million to $4.24 million, making it the highest average annual total cost ever reported in this report. Therefore, it is necessary to solve the supervision of the smart aging platform from the perspective of privacy protection, aiming to reduce the privacy worries of the elderly and enhance the supervision of the smart aging platforms.

Currently, there are still some problems that need to be solved urgently in various area in the process of supervision of the smart elderly care platform. Older people (the elderly) still have a low level of participation in the supervision of the wisdom aging platform [4], and the low level of privacy protection of the platform service provider will lead to the improper operation of the wisdom aging platform and directly affect the elderly’s willingness to participate in the supervision of the platform. The platform service providers themselves have the behavior of rational people seeking to maximize profits [5], but improper government regulation will lead to fraudulent subsidies and the illusion of formal “high-quality protection” of older people’s privacy. The government strongly advocates the development of smart aging platforms, meets the service interests and requirements of the elderly, standardizes the platform, and formulates regulatory measures [6]. However, from a practical point of view, the government will be influenced by its own interest preferences, and will put more emphasis on the regulatory investment of smart aging platforms. For example, platform service providers actively increase investment to improve the level of platform privacy protection, allowing the government a “free-ride” to reduce regulatory investment [7].

The current research on personal privacy disclosure of platforms mainly adopts the statistical methods of questionnaire survey and hypothesis test [8], and the survey targets the elderly. However, the supervision of smart aging platforms is not only related to the privacy concerns and income perception of the elderly, but also influenced by the willingness of the elderly to participate in supervision, the level of privacy protection of platform service providers and the degree of government supervision. Evolutionary games, as a research method under the conditions of multiple subjects, limited rationality, long term, and information asymmetry, can analyze the dynamic trends in factors, such as benefits, costs, and losses, on the behavior patterns of subjects under the privacy leakage behavior. Based on this, this paper will introduce evolutionary game theory from the perspective of privacy protection to discuss the supervision strategy of smart aging platforms. In addition to considering the privacy concerns and benefit perceptions of the elderly, the willingness of the elderly to participate in supervision, the level of privacy protection of platform service providers and the degree of government supervision also play an important role in the research into the supervision of smart aging platforms. Assuming that the elderly, platform service providers and the government are the main players in the game, it is possible to solve the evolutionary stability strategy, and explore the influence of different factors, such as benefits, costs, losses, rewards, and punishments, on the regulatory strategy of smart aging platforms, and put forward targeted suggestions for the healthy development of smart aging service field.

The rest of this paper is organized as follows. The second section reviews the literature, analyzes the wisdom aging platform and privacy research conducted by predecessors, analyzes its limitations based on the existing research results, and determines the research methods, contents, and research objectives of this paper. The third section introduces interest demands and subject relationships among the elderly, platform service providers, and the government. The fourth section analyzes stability, determines the evolutionary stability strategy of unilateral and tripartite systems, and identifies the key factors that affect the decision-making behavior of tripartite entities. In the fifth section, simulation analysis and system optimization are performed using MATLAB. The specific content is the initial state simulation, parameter change simulation, and optimal state simulation. The sixth section puts forward conclusions and suggestions according to the research results.

## 2. Literature Review

Emerging technologies such as AI, big data, and mobile communication are widely used in the field of aged care services and are promoting the rapid development of a series of “smart elderly care” products such as elderly care services and apps. At present, the development, design, and evaluation of smart aging platforms, privacy protection technology, and privacy research into smart aging platforms are attracting the attention of many experts and scholars.

### 2.1. The Development and Use of Smart Aging Platforms

From the perspective of the development and use of a smart aging platform, Xia et al. [9] evaluated an old-age health check-up app by a comprehensive evaluation method combining analytic hierarchy process and fuzzy theory, and pointed out that in the evaluation research of old-age health check-up apps, the operation learning and information processing of software are the most important. Massollar et al. [10] put forward a tracking app for the elderly to reduce exposure to health risks against the background of legal issues of personal privacy and the novel coronavirus pandemic and conducted a technical evaluation and simulated data experiments on the app. The results verified the effectiveness of the tracking app for the elderly in preventing infectious risks. Lee et al. [11] designed a Japanese elderly opportunity matching app named “GBER”, which could collect new insights on the perception, behavior, daily life, and use of smartphones by the elderly, as well as their impression and ability to use GBER through user tests and semi-structured interviews. Zhao et al. [12] designed an app for the elderly to live and keep healthy in combination with their experiences, and studied the influencing factors such as physiological characteristics, psychological characteristics, and cultural characteristics of the elderly, and found out the difficulties in designing the old-age app. The research results showed that an old-age app should have the characteristics of simplicity, visualization, one-click help, infallibility, and auxiliary memory. Cristiano et al. [13] interviewed 24 older people (aged > 65 years) and 118 clinicians through focus group activities and a web-based questionnaire to understand the requirements of the Smart Health Platform. The key features of the smart health platform were designed to address these requirements. With the My Active and Health Aging platform, Rainero et al. [14] evaluated the effectiveness of a tailored multi-disciplinary intervention to prevent quality of life decline in older people.

### 2.2. Privacy Protection Technology Research

In terms of privacy protection technology research, which is extensive in mobile communication and big data environments, Prodanoff et al. [15] describe a standards-based security framework for documenting security and privacy vulnerabilities found in mHealth Apps, to facilitate regulatory and standardized updates and threat mitigation and make it easier for patients to understand the impact of specific threats on their privacy and security implications. Small et al. [16] developed a platform called “Action ADE” to solve the privacy leakage caused by ICT, which was able to record and exchange patient-specific antibody-dependent enhancement (ADE) information, and discuss privacy issues from the patient’s point of view, and put forward three design suggestions to balance the needs of sharing and protecting personal health information. Lai et al. [17] proposed an access control model of hierarchical broadcast encryption according to the privacy access control requirements of intelligent service systems. Compared with the traditional centralized authorization strategy, this model can better prevent leakage of users privacy. Awan et al. [18] proposed a big data tool (Apache Spark) to detect DDoS attacks on sensitive data, and compared it with non-big data methods to demonstrate the superiority of Apache Spark and improve the security of private data.

### 2.3. Privacy Research for Smart Aging Platforms

From the perspective of privacy research of smart aging platforms, Elahi et al. [19] contacted Android expert users, learned about their privacy decisions and preferences, and used them as a reference to improve the privacy protection of elderly Android users in smart cities. Two algorithms, PPPA-1 and PPPA-2, were proposed to protect the privacy of elderly Android users in smart cities and realized cognitive unloading by removing user interaction needs. Quan-Haase et al. [20] conducted in-depth interviews with 40 elderly people living in East New York, Toronto, Canada, to investigate their concerns about online privacy. The study found that safety and privacy are top concerns for older people in New York, and 40% percent chose not to subscribe to certain websites or online services. Li et al. [21] developed an acceptance model of a smart wearable system for the elderly by using a structural equation model and collected 146 survey samples from the elderly over 60 years old to verify the model. The research results show that performance risks such as privacy issues of the smart wearable system will significantly reduce the perceived usefulness of the system, thus leading to a decrease in the acceptance of the smart wearable system by the elderly. Liu [22] realized the structural equation model of privacy and other related factors on the willingness to accept smart aging care services based on collecting and sorting out relevant data. Regression results show that privacy concerns have no significant positive impact on the willingness to accept smart aged care services. The possible explanation for this situation is that privacy concerns may, to a certain extent, affect the adoption of smart aging care services by some elderly people.

### 2.4. Research on the Application of Evolutionary Game Theory

Evolutionary game theory is also widely used in the study of privacy disclosure and privacy supervision. He et al. [23] developed an evolutionary game model to analyze the strategic behavior of two types of players in the provision of elderly services, the government sector, and the private sector. Yue et al. [24] focused on the impact of penalties and operating subsidies on the evolutionary stabilization strategies of the two players, and pointed out that appropriate increases in penalties and operating subsidies are conducive to the regulation of service quality in retirement PPP projects. Zhu et al. [25] analyzed the tripartite game model among patients, medical software vendors, and the government through an evolutionary game, and showed that the loss of privacy disclosure harmed the willingness to use medical software. Wang et al. [26] analyzed the dynamic strategy changes in government, app stores, and app software under government privacy control by using the tripartite evolutionary game and prospect theory. The research results show that the sensitivity of government perceived value and loss avoidance are the key influencing factors. Zhou et al. [27] built a game model between over-collecting apps and the government through the evolutionary game method, studied the influencing factors of over-collecting app personal data protection and use strategy selection, and found that over-collecting app and government strategy selection are related to data leakage probability. From the standpoint of privacy protection, Han et al. [28] developed a tripartite evolutionary game model involving patients, medical service institutions, and the government. The conclusion demonstrates that by examining the impact of factors such as investment, income, and cost on privacy protection and willingness to share medical data, the level of privacy protection of medical data can be improved, and the process of sharing medical data in China can be promoted.

Based on the existing research results, it can be seen that the research on smart aging platforms and privacy mainly focuses on technology implementation and application analysis. However, in the process of cloud computing privacy protection, it is difficult to choose the optimal strategy [29], and participants often weigh their benefits, costs, and losses from an economic perspective and make decisions. At the same time, the privacy research objects of smart aging platforms focus mainly on the elderly, lacking the privacy protection behavior analysis of platform service providers and government supervision behavior analysis, and ignoring the long-term nature of smart aging platform supervision and privacy leakage, and the actors should choose the best decision in the dynamic process. In view of the above analysis, this paper will build a tripartite evolutionary game model, study the regulatory strategy of the smart aging platform under the background of privacy protection, analyze the evolutionary stability strategy of the tripartite agents and the key factors affecting the agent strategy selection, and simulate and optimize the evolutionary path of the behavior law of the tripartite system through MATLAB numerical simulation.

The following are the main contributions of our study:First, it constructs a three-party evolutionary game model under the perspective of privacy protection and measures the changes in benefits, costs, and losses of the three parties from the perspective of economics, to make decisions.Second, in addition to analyzing the willingness of the elderly to participate in the regulation of the smart aging platform, an analysis of the privacy protection behavior of the platform service providers and an analysis of the government’s regulatory behavior are also conducted.Third, the stability analysis of the unilateral evolution strategy is not only performed using the replicated dynamic equations, but also the transformation of the unstable point to the stable point is analyzed by Lyapunov theory.

## 3. Model Building

### 3.1. Application Scenario

The participants involved in the smart aging platform include not only elderly individuals, but also many new subjects such as platform service providers, communication operators, and third-party regulators [30]. This paper will examine the issue of regulation of smart aging platforms from the perspective of privacy breaches. For the elderly, they can act as supervisors, pursuing reasonable prices for the operation and services of the wisdom aging platform, diversification of service types, as well as focusing on the experience of the platform services, which are the main sources of revenue for the platform [31]. As beneficiaries of the wisdom aging platform, their interest is to bring convenience and spiritual and emotional comfort to their lives through the aging platform’s services. For the platform service providers, the main task is to integrate social quality resources for the elderly groups and to match the elderly people’s own needs for elderly care services in a timely manner. As the core stakeholder of the platform, their main interest is to gain better influence and corresponding economic benefits through the platform [32]. From the government’s perspective, it provides financial support for the implementation of the smart aging platform on the one hand and supervises the security of the smart aging platform on the other. Its interests are to develop regulatory measures to improve the level of privacy protection of the smart senior care platform [33], to meet the senior care needs of many elderly people, and to enable the level of government credibility to be improved. Furthermore, as platform service providers become more aware of privacy protection and actively choose to protect the privacy of older people at a high-quality level, the government may have a “free-ride” mentality and choose to under-invest in regulation.

Given the long-term nature and ongoing nature of privacy protection [34,35], the information asymmetry between older people, platform providers, and the government, and the uncertainty about the effectiveness of regulation, the game behavior of the three parties is characterized by limited rationality. It is difficult for game players to seek the ideal strategy under this characteristic, but they must seek the optimal option by replicating dynamic processes, such as learning, in several games [36]. Therefore, the elderly, the platform service provider, and the government are defined as the three parties of the game, and the subject relationship is shown in Figure 1.

### 3.2. Model Assumptions and Construction

The following hypotheses are presented to allow for a clearer analysis of the game model and for an in-depth study of the strategic choices of the subjects of the three-way game, as well as the impact of each component on changes in the subjects’ strategic behavior.

**Hypothesis** **1** **(H1).**
*“Economic man” hypothesis. When the elderly use smart aging platform services, they pursue the maximization of the benefits of aging services; Other operators of platform service providers are similar, pursuing profit maximization; As an advocate and promoter of the development of smart aging platform services, the government pursues the maximization of government credibility and social benefits [37].*


**Hypothesis** **2** **(H2).**
*Limited rational hypothesis. Due to the long-term nature of privacy disclosure, the asymmetry of information among the elderly, the platform service provider, and the government, and the uncertainty of supervision effect, constant dynamic adjustment of the game strategy as the best decision is necessary. Therefore, the assumption that the subject of the game is bounded rationality is more in line with the practical application situation of the smart aging platform.*


**Hypothesis** **3** **(H3).**
*Hypothesis of strategy selection. There are two strategies of “participation” and “non-participation” for the elderly to participate in the supervision of the smart aging platform, and the parameter x(0≤x≤1)*
*shows the participation of the elderly in platform supervision. To reflect the privacy leakage of the platform, platform service providers have two strategies—high-quality protection and low-quality protection—for the privacy of the elderly, with the parameter y(0≤y≤1)*
*reflecting the platform’s level of protection for elderly privacy. In order to promote more elderly people to use smart aging platform services and enhance the credibility of the government, the government will adopt two strategies of “high investment supervision” and “low investment supervision” with the parameter z(0≤z≤1)*
*reflecting the degree of supervision of the smart aging platform’s privacy protection level.*


**Hypothesis** **4** **(H4).**
*Hypothesis of model parameters. To express the tripartite benefit matrix conveniently and ensure the objectivity of parameter setting as much as possible, after referring to relevant literature [8,26,38] and consulting experts’ opinions, the parameters are set and defined from three angles of cost, benefit, and loss in combination with the actual privacy leakage situation, as shown in Table 1.*


The tripartite subject returns are calculated under various strategy combinations using the above-mentioned assumptions and parameter settings, and the game subject returns matrix is built, as shown in Table 2.

## 4. Stability Analysis

### 4.1. Stability Analysis of Unilateral Evolutionary Strategies

(1)Stability analysis of elderly evolutionary strategies

Assuming that the elderly chooses “participation” and “non-participation” to supervise the smart aging platform, the expected benefits are respectively $E_{11}$ and $E_{12}$, the average benefit is ${\bar{E}}_{1}$, then:(1)$$E_{11}=y(\beta -\alpha )L_{P}+zR_{t}+R_{P}-\beta L_{P}$$
(2)$$E_{12}=yzR_{t}+(1-y)zR_{t}=zR_{t}$$
(3)$${\bar{E}}_{1}=xE_{11}+(1-x)E_{12}$$

The replication dynamic equation for the elderly evolutionary strategy is:(4)$$f(x)=\frac{dx}{dt}=x(E_{11}-{\bar{E}}_{1})=x(1-x)[y(\beta -\alpha )L_{P}+R_{P}-\beta L_{P}]$$

When $y=\frac{\beta L_{P}-R_{P}}{(\beta -\alpha )L_{P}}$, f(x)≡0, when all values of x taken are steady state and both strategies of the elderly are the ESS. When $y>\frac{\beta L_{P}-R_{P}}{(\beta -\alpha )L_{P}}$, let f(x)=0 to get the equilibrium points x=0 and x=1, then we have $f^{\prime }(0)>0$, $f^{\prime }(1)<0$, x=1 is the stable point, and participation is the ESS. When $y<\frac{\beta L_{P}-R_{P}}{(\beta -\alpha )L_{P}}$, let f(x)=0 to get the equilibrium points x=0 and x=1, then there are $f^{\prime }(0)<0$, $f^{\prime }(1)>0$, then x=0 is the stable point, and non-participation is the ESS. The elderly dynamic trend phase diagram is shown in Figure 2.

As can be seen from Figure 2, if the loss of privacy is greater than the gain in service, $s_{2}$ becomes larger and the shaded cross-section moves in the positive direction of the *y*-axis, the elderly tend not to participate in the strategy. If the loss of privacy is smaller than the gain of service, the $s_{1}$ space becomes larger and the shaded cross-section shifts to the negative direction of *y*-axis, the elderly tend to participate in the strategy. In addition, as $(\beta -\alpha )L_{P}$ decreases, the $s_{2}$ space becomes larger, and the elderly still tend to not participate.

(2)Stability analysis of the evolutionary strategy of the platform service providers

Let the expected benefits of high-quality protection and low-quality protection of public privacy in the platform service providers be $E_{21}$ and $E_{22}$, respectively, and the average benefit ${\bar{E}}_{2}$, then:(5)$$E_{21}=x[R_{A}+(1-\alpha )r_{A}-\beta L_{A}+C_{1}]+zR_{e}-C_{1}$$
(6)$$E_{22}=x[R_{A}+(1-\beta )r_{A}-\beta L_{A}]+z[(1-\lambda )R_{e}-\lambda F_{s}]-C_{2}$$
(7)$${\bar{E}}_{2}=yE_{21}+(1-y)E_{22}$$

The replication dynamic equation for the evolutionary strategy of the platform service providers is:(8)$$f(y)=\frac{dy}{dt}=y(E_{21}-{\bar{E}}_{2})=y(1-y)[x(\beta -\alpha )(r_{A}+L_{A})+z\lambda (R_{e}+F_{s})+C_{2}-C_{1}+xC_{1}]$$

When $z=\frac{C_{1}-C_{2}-x[(\beta -\alpha )(r_{A}+L_{A})+C_{1}]}{\lambda (R_{e}+F_{s})}$, f(y)≡0, when all values of y taken are steady state and both strategies of the platform service providers are ESS. When $z>\frac{C_{1}-C_{2}-x[(\beta -\alpha )(r_{A}+L_{A})+C_{1}]}{\lambda (R_{e}+F_{s})}$, let f(y)=0 to get the equilibrium points y=0 and y=1, then we have $f^{\prime }(0)>0$, $f^{\prime }(1)<0$, then y=1 is the stable point, and high-quality protection is the ESS. When $z<\frac{C_{1}-C_{2}-x[(\beta -\alpha )(r_{A}+L_{A})+C_{1}]}{\lambda (R_{e}+F_{s})}$, let f(y)=0 to get the equilibrium points y=0 and y=1, then there are $f^{\prime }(0)<0$, $f^{\prime }(1)>0$, then y=0 is the stable point, and low-quality protection is the ESS. The dynamic trend phase diagram of the platform service providers is shown in Figure 3.

As can be seen from Figure 3, if other parameters remain unchanged and the input cost of the platform service provider decreases, the shaded cross-section shifts to the lower left, the $s_{3}$ space increases, and the platform service provider tends to the high-quality protection strategy. Similarly, the higher the level of government incentives and penalties, the more the platform service providers choose the high-quality protection strategy. When the loss of privacy leakage and future revenue become larger, the platform service providers tend to choose the high-quality protection strategy. When the β−α difference becomes smaller, the $s_{4}$ space becomes larger, and the platform service provider chooses the low-quality protection strategy.

(3)Analysis of the stability of the government’s evolutionary strategy

Assuming that the privacy protection level of the government’s high input supervision and low input supervision platforms is $E_{31}$ and $E_{32}$, respectively, and the average income is ${\bar{E}}_{3}$, the same can be obtained:(9)$$E_{31}=xy(R_{G}+L_{G}+\beta T-\alpha T)-x(L_{G}+\beta T)+(1-y)\lambda (R_{e}+F_{s})-C_{G}-R_{e}$$
(10)$$E_{32}=xy(R_{G}+L_{G}+\beta T-\alpha T)-x(L_{G}+\beta T)$$
(11)$${\bar{E}}_{3}=zE_{31}+(1-z)E_{32}$$

The replication dynamic equation for the evolutionary strategy of the government is:(12)$$f(z)=\frac{dz}{dt}=z(E_{31}-{\bar{E}}_{3})=z(1-z)[\lambda (R_{e}+F_{s})-y\lambda (R_{e}+F_{s})-C_{G}-R_{e}]$$

When $y=1-\frac{C_{G}+R_{e}}{\lambda (R_{e}+F_{s})}$, f(z)≡0, when all values of z taken are steady state and both strategies of the government are the ESS. When $y>1-\frac{C_{G}+R_{e}}{\lambda (R_{e}+F_{s})}$, let f(z)=0 to get the equilibrium points z=0 and z=1, then we have $f^{\prime }(0)<0$, $f^{\prime }(1)>0$; then z=0 is the stable point, and high input supervision is the ESS. When $y<1-\frac{C_{G}+R_{e}}{\lambda (R_{e}+F_{s})}$, let f(z)=0 to get the equilibrium points z=0 and z=1, then $f^{\prime }(0)>0$, $f^{\prime }(1)<0$, and z=1 is the stable point, and low input supervision is the ESS. The government dynamic trend phase diagram is shown in Figure 4.

As can be seen from Figure 4, the space of $s_{5}$ and $s_{6}$ in the initial strategy space is related to the cost of government regulation, and s6 becomes larger when the cost of regulation increases, which tends towards low input regulation. When the government chooses the strategy of high input regulation, it is found that there is a higher the probability that the platform service provider chooses low-quality protection; the higher the fine, the more government tends to choose high input regulation. The government tends to prefer high input regulation.

### 4.2. Stability Analysis of the Evolutionary Strategy of the Tripartite System

A stability study of the evolutionary stabilization strategy of the tripartite system will be performed for further evaluation of the ideal evolutionary stabilization strategy of the tripartite system and the main elements impacting the behavioral patterns of the tripartite subjects.

(1)Determination of the point of progressive stability

A tripartite evolutionary dynamical system consisting of replicated dynamic equations for every single party:(13)$$f(x)=x(1-x)[y(\beta -\alpha )L_{P}+R_{P}-\beta L_{P}]=0$$
(14)$$f(y)=y(1-y)[x(\beta -\alpha )(r_{A}+L_{A})+z\lambda (R_{e}+F_{s})+C_{2}-C_{1}+xC_{1}]=0$$
(15)$$f(z)=z(1-z)[\lambda (R_{e}+F_{s})-y\lambda (R_{e}+F_{s})-C_{G}-R_{e}]=0$$

From Equations (13)–(15), the dynamical system is solved in conjunction to obtain the system equilibrium points $A_{1}(0,0,1)$, $A_{2}(0,1,1)$, $A_{3}(1,0,1)$, $A_{4}(1,1,1)$, $A_{5}(0,0,0)$, $A_{6}(0,1,0)$, $A_{7}(1,0,0)$, $A_{8}(1,1,0)$, and $A_{9}(x^{\prime },y^{\prime },z^{\prime })$. It follows from the probabilities in the model assumptions that $y^{\prime }\in [0,1]$, so $\beta L_{P}-R_{P}\leq (\beta -\alpha )L_{P}$.

In the dynamic replication system of multi-subject evolutionary game, strategy X is asymptotically stable in the dynamic replication system of the multi-group evolutionary game when and only when X is a strict Nash equilibrium [39], so only the asymptotic stability of the equilibrium points of the system involving pure strategies need to be analyzed, and the equilibrium points of the system satisfying the condition include eight points from $A_{1}$ to $A_{8}$. According to the Lyapunov system stability discriminant, when all the eigenvalues of the Jacobi matrix are less than zero, the equilibrium point is asymptotically stable [40]; when at least one eigenvalue in the Jacobi matrix is greater than zero, the equilibrium point is not asymptotically stable. Denote the Jacobi matrix as W:(16)$$W=\left[\begin{array}{ccc} \frac{\partial f(x)}{\partial x} & \frac{\partial f(x)}{\partial y} & \frac{\partial f(x)}{\partial z}\\ \frac{\partial f(y)}{\partial x} & \frac{\partial f(y)}{\partial y} & \frac{\partial f(y)}{\partial z}\\ \frac{\partial f(z)}{\partial x} & \frac{\partial f(z)}{\partial y} & \frac{\partial f(z)}{\partial z} \end{array}\right]$$

The eigenvalues corresponding to each of the eight system equilibrium points may be found in Table 3 by substituting each of the partial derivative function equations in the matrix of Equation (16).

Combining the parameter sizes in the model assumptions reveals that $\beta L_{P}-R_{P}\leq (\beta -\alpha )L_{P}$, $C_{G}+R_{e}>0$, $(\beta -\alpha )(r_{A}+L_{A})+\lambda (R_{e}+F_{s})+C_{2}>0$, and $(\beta -\alpha )(r_{A}+L_{A})+C_{2}>0$. Thus, $A_{2}(0,1,1)$, $A_{3}(1,0,1)$, $A_{4}(1,1,1)$, $A_{6}(0,1,0)$, and $A_{7}(1,0,0)$ are not asymptotically stable points.

The usage of an aging app by elderly families can solve difficulties such as mobility issues, medical services for the elderly, and emergency support, all of which are important for the healthy development of aged services. The platform service providers’ high-quality protection of the privacy of elderly families and their families may enhance their willingness to participate in the supervision of the smart aging platforms. The government may adopt a free-riding mentality to save money. With the improvement of platform service providers’ awareness of privacy protection, they actively adopt the strategy of high-quality protection. To lower the expense of regulation, the government can pick low input regulation. We can select $A_{8}(1,1,0)$ as the optimal asymptotic stabilization point, i.e., (participation, high-quality protection, low input regulation) as the best ESS, because only the three eigenvalues are negative. The three-party game space is:$$S=s_{1}\cap s_{3}\cap s_{6}=\left\{(x,y,z)|y>\max\{1-\frac{C_{G}+R_{e}}{\lambda (R_{e}+F_{s})},\frac{\beta L_{P}-R_{P}}{(\beta -\alpha )L_{P}}\},z>\frac{C_{1}-C_{2}-x[(\beta -\alpha )(r_{A}+L_{A})+C_{1}]}{\lambda (R_{e}+F_{s})}\right\}$$

This is a more perfect equilibrium state for the smart aging platform’s stable development, and it is the optimal strategic goal we are pursuing. As a result, $A_{8}(1,1,0)$ serves as a reference target for examining the asymptotic stability of $A_{1}(0,0,1)$ and $A_{5}(0,0,0)$.

(2)Stability analysis of $A_{1}(0,0,1)$

When $R_{P}<\beta L_{P}$, $\lambda (F_{s}+R_{e})<C_{1}-C_{2}$, and $C_{G}+(1-\lambda )R_{e}<\lambda F_{s}$, $A_{1}(0,0,1)$ is the asymptotic stability point and the ESS is non-participation, low-quality protection, high input regulation. From $R_{P}<\beta L_{P}$, it is clear that if the service benefits received by the elderly are less than the loss caused by privacy leakage, the elderly will choose the non-participation strategy. Therefore, platform service providers should strengthen the level of privacy protection for the elderly, and at the same time expand the service income for the elderly, so as to encourage the elderly to participate in the supervision of the smart aging platform. According to $\lambda (F_{s}+R_{e})<C_{1}-C_{2}$, the sum of government subsidies and fines for platform service providers is less than the difference between the platform’s input cost when there is high-quality protection and low-quality protection, and the platform service providers tend to low-quality protection of user privacy in the case of high input regulation by the government. The government can raise the subsidy for high-quality protection while lowering the penalty for low-quality protection, or platform service providers can expand the input cost of low-quality protection and reduce the input cost of high-quality protection to promote high-quality protection of the privacy of elderly home users. The analysis of $C_{G}+(1-\lambda )R_{e}<\lambda F_{s}$ shows that if the sum of the cost of regulation and the subsidy given to the senior care app platform is less than the fine received from the platform, the government will choose the high input regulation strategy. However, to save money, the government frequently employs the strategy of low input regulation. If the amount of subsidy is increased, it will encourage the platform service providers to take the initiative to provide high-quality protection of the privacy of elderly families and their family users, thereby enhancing the government’s credibility. An increase in the number of subsidies can benefit the elderly, the platform service providers, and the government.

(3)Stability analysis of $A_{5}(0,0,0)$

When $R_{P}<\beta L_{P}$, $C_{2}<C_{1}$ and $C_{G}+(1-\lambda )R_{e}>\lambda F_{s}$, $A_{5}(0,0,0)$ is the asymptotic stability point and ESS is non-participation, low-quality protection, low input regulation. If $R_{P}<\beta L_{P}$ is similar to the stability analysis of $A_{1}(0,0,1)$, the platform service providers should increase the level of privacy protection for public users while also expanding the service benefits obtained by the elderly, to encourage the elderly to participate in supervising the smart aging platform. If $C_{2}<C_{1}$, i.e., the investment cost of low-quality protection is less than that of high-quality protection, the platform service providers will choose low-quality protection. To enable the platform service providers to choose the strategy of high-quality protection, the input cost $C_{2}$ should be expanded or reduced $C_{1}$, so that the difference between the input cost of the service providers when choosing the two strategies is not significant as far as possible. When we look at $C_{G}+(1-\lambda )R_{e}>\lambda F_{s}$, we can see that if the government actions high input projects, we obtain high input results. If the cost of regulation plus the subsidies granted to the platform service providers exceeds the penalty earned from the platform, the government will opt for low input regulation, which is exactly what the government expects.

## 5. Simulation Analysis and System Optimization

MATLAB was used to simulate the evolutionary game process of the three-party system and numerical simulations were carried out for $A_{8}(1,1,0)$, $A_{1}(0,0,1)$, and $A_{5}(0,0,0)$ to represent the evolutionary game behavior of the three-party system more clearly and to test the correctness of the game model.

### 5.1. Numerical Simulation Analysis of $A_{8}(1,1,0)$

The assignment is simulated in conjunction with the realistic situation according to the asymptotic stability requirement of $A_{8}(1,1,0)$, and the assignment is shown in Table 4, with the results acquired by the simulation software as shown in Figure 5.

Combining the constraints in Table 3, we can see that $\alpha L_{P}\leq R_{P}$, the service income obtained by the elderly is greater than the loss caused by privacy leakage. At this time, the elderly will participate in the supervision of the smart aging platform; $-[(\beta -\alpha )(r_{A}+L_{A})+C_{2}]<0$ is constant, and the platform service providers always choose to protect the public’s privacy with high quality; $-C_{G}-R_{e}<0$ is also constant and holds, indicating that the government would stick to its low input regulation policy. The three parties’ behavioral patterns are in a perfect equilibrium state (see Figure 5), i.e., they are pursuing the best strategic goal.

### 5.2. Numerical Simulation Analysis of $A_{1}(0,0,1)$

According to the asymptotic stability constraint of $A_{1}(0,0,1)$, the assignment is simulated in conjunction with the realistic situation; the assignment is shown in Table 5, and the results are obtained through the simulation program as shown in Figure 6.

The system equilibrium point $A_{1}(0,0,1)$ (as shown in Figure 6) stabilizes the initial dynamic evolutionary trend, indicating non-participation, low-quality protection, high input regulation as the ESS. To bring the equilibrium closer to the optimal strategic goal, platform service providers should strengthen the level of privacy protection for the elderly, and at the same time expand the service income for the elderly, so as to encourage the elderly to participate in the supervision of the smart aging platform. The government can promote high-quality protection of the privacy of the elderly family users by increasing subsidies for high-quality protection and fines for low-quality protection, or by expanding and reducing the input cost of low-quality protection by platform service providers. To achieve the goal of free-riding, the government can increase the amount of subsidy, opting for the low input regulation option. Combined with the stability analysis of $A_{1}(0,0,1)$, we can see from Figure 7 and Figure 8 that the three-party system converges to (1,1,0) as parameter $R_{P}$ increases and parameter $L_{P}$ decreases, suggesting that the remaining two parties are potentially influenced by the decision of older people to choose high-quality protection and high input regulation strategies, respectively, when they are involved in supervision. The remaining two parties are potentially influenced by the older person’s decision to choose the high-quality protection and high input regulation strategies, respectively. In Figure 9, however, changes in the parameters $R_{e}$, $F_{s}$, $C_{1}$, and $C_{2}$ cause the three-way system to converge on (0,1,0). Clearly, both the platform provider and the government have achieved their objectives, while the older person has not changed, possibly because the change in the relevant parameters does not involve a loss or gain for the older person. In order to optimize the system, the above six key parameters are changed at the same time, the values of $F_{s}$, $R_{P}$, $R_{e}$, $L_{P}$, $C_{1}$, and $C_{2}$ are modified from (9,4,1,10,6,1) to (15,10,3,0,1,6), but the rest of the parameters stay intact. The simulation results, shown in Figure 10, clearly show that ESS is transformed from the original (non-participation, low-quality protection, high input regulation) to (participation, high-quality protection, low input regulation), resulting in the game system’s optimization.

### 5.3. Numerical Simulation Analysis of $A_{5}(0,0,0)$

The allocation is simulated in association with the practical scenario according to the asymptotic stability requirement of $A_{5}(0,0,0)$; the allocation is shown in Table 6, and the results are generated by the simulation program as shown in Figure 11.

The initial dynamic evolutionary trend stabilizes at the system equilibrium point $A_{5}(0,0,0)$ (e.g., Figure 11), indicating (non-participation, low-quality protection, low input regulation) as ESS. Through the foregoing analysis, to make the equilibrium state closer to the optimal strategic goal, platform service providers should strengthen the level of privacy protection for the elderly, and at the same time expand the service income for the elderly, to encourage the elderly to participate in the supervision of the smart aging platform. Platform service providers should expand the input cost or reduce the input cost, to make the input cost difference between the two strategies as insignificant as possible, and thus tend to choose high-quality protection. The government can appropriately increase the number of subsidies for the platform service providers to achieve the low input regulation strategy. Combined with the stability analysis of $A_{5}(0,0,0)$, we can see from Figure 12 and Figure 13 that the three-party system converges to (1,1,0) as parameter $R_{P}$ increases and parameter $L_{P}$ decreases, suggesting that the remaining two parties are potentially influenced by the decision of older people to choose the high-quality protection and high input regulation strategies, respectively, when they are involved in supervision. The remaining two parties are potentially influenced by the older person’s decision to choose the high-quality protection and high input regulation strategies, respectively. In Figure 14, however, changes in the parameters $R_{e}$, $F_{s}$, $C_{1}$, and $C_{2}$ cause the three-way system to converge on (0,1,0). Clearly, both the platform provider and the government have achieved their objectives, while the older person has not changed, possibly because the change in the relevant parameters does not involve a loss or gain for the older person. Therefore, in order to optimize the system, the above six key parameters are changed at the same time, the values of $R_{P}$, $L_{P}$, $R_{e}$, $F_{s}$, $C_{1}$, and $C_{2}$ are transformed from 4,10,3,9,6,1 to (7,5,6,0,5,6); the simulation results are shown in Figure 15, at which point the ESS is transformed from (non-participation, low-quality protection, low input regulation) to (participation, high-quality protection, low input regulation), and the gaming system reaches optimization.

## 6. Conclusions and Suggestions

Based on the characteristics of limited rationality and information asymmetry, this paper applies evolutionary game theory to the supervision research of smart aging platforms, establishes a tripartite evolutionary game model with the elderly, the platform service providers, and the government as the main bodies, and analyzes the stability of evolutionary strategies of unilateral and tripartite systems by copying the dynamic equations obtained from the model, and verifies the effectiveness of the evolutionary game model according to the simulation analysis. The final results: (1) $A_{8}(1,1,0)$, $A_{1}(0,0,1)$, and $A_{5}(0,0,0)$ are all asymptotically stable, with the ideal equilibrium state being $A_{8}(1,1,0)$; (2) The elderly will actively participate in supervising the smart aging platform if the benefits of the service they receive outweigh the losses caused by the privacy breach. In the case of high investment supervision by the government, the sum of government subsidies and fines for platform service providers is greater than the difference between the input costs of high-quality protection and low-quality protection, and platform service providers tend to high-quality protection and the privacy of the elderly. The input cost of platform service providers choosing low-quality protection is greater than that of high-quality protection, and the platform will also choose high-quality protection when there is no significant difference between the two strategies. When the sum of the supervision cost and the subsidy to the platform service providers when the government conducts high input supervision is greater than the fine, the government will choose low input supervision; (3) Both $A_{1}(0,0,1)$ and $A_{5}(0,0,0)$ achieve the desired equilibrium after the parameters are adjusted to optimize the gaming system.

A summary of the research findings concludes that the optimal ESS is participation, high-quality protection, low input regulation, i.e., the optimal strategic goal; the key factors influencing the behavioral patterns of the three parties are the benefit of the service received by the elderly, the loss of privacy leakage, the input cost of the platform service providers, and the number of government rewards and penalties, respectively. In conjunction with the above research findings, to enhance the enthusiasm of the elderly to participate in the supervision of the smart aging platform and strengthen the privacy protection level of the smart aging platform, the following suggestions are put forward:

First, increase the elderly’s schedule flexibility when using the platform service providers. For example, if the public can schedule appointments on the smart aging platform and receive senior care services at specific times, increasing the schedule flexibility between the public and the platform service providers will reduce the waiting time of elderly families and their families, improving the quality of services to the elderly and making the benefits of services received from the platform more significant, thus enhancing the enthusiasm of the elderly to participate in the supervision of smart aging platforms.

Second, reduce the input cost of platform service providers in choosing high-quality protection privacy. The input cost of privacy protection is one of the important influencing factors for platform service providers to choose game strategies. If the input cost of high-quality protection privacy is too high, platform service providers will tend to choose a low-quality protection strategy, which will affect the elderly’s willingness to participate in the supervision of smart aging platforms, resulting in a lose–lose situation that the elderly refuse to participate in the supervision of smart aging platforms and platform service providers are unwilling to protect privacy. When platform service providers choose high-quality protection and low-quality protection, the input cost difference is not significant, and they will still choose the high-quality protection strategy. Therefore, we can adapt the method of narrowing the input cost gap between the two strategies, and raise or limit the lower limit of low-quality protection input by formulating laws and regulations. If the input cost difference between the two strategies is extremely small, platform service providers are more willing to provide high-quality protection of public privacy, and the willingness to participate in the supervision of smart aging platforms for the elderly is stronger.

Third, increase the number of government awards and penalties for platform service providers appropriately. The government should give incentives and punishments for platform service providers that provide high-quality protection of public privacy. The platform service provider will generally escape penalties and earn government subsidies if it opts for a high-quality protection plan, but a low-quality protection strategy will increase the danger of privacy breaches for the elderly, posing a risk to the government. The government would be responsible for the risk of a penalty and the payment of a fine. The degree of incentives and sanctions, however, must be proportionate. If too much is subsidized, platform service providers will reach a “sweet spot” and give the impression of formalistic high-quality protection to profit from the government; if too much is penalized, the government would see it as high investment regulation. If the punishment is excessive, it amounts to high input regulation in the eyes of the government, which does not achieve the goal of free-riding and is not consistent with the expected decision; thus, the government should set an appropriate number of rewards and penalties.

Lastly, enhance the technology and management of the smart aging platforms in terms of privacy and confidentiality. The identification and assessment of smart aging platforms privacy leakage issues must be strengthened, and the “hard bone” must be gnawed from both technical and management perspectives. In terms of technology, we should develop and publish software development kits and mobile phone operating systems to analyze the security of information for the elderly, and perform in-depth assessments of senior citizens to decrease the damage caused by privacy leakage. In terms of management, we should improve the detection, exposure, and punishment of illegal gathering and use of information from elderly individuals and their families, and interview, warn, shelve, and fine them following the law. To effectively deter privacy exposure, we will increase our efforts to discover, expose, and punish illegal acquisition and use of information about older families and their families, and we will interview, warn, take down, and apply fines under the law.

## Figures and Tables

**Figure 1 ijerph-19-05778-f001:**
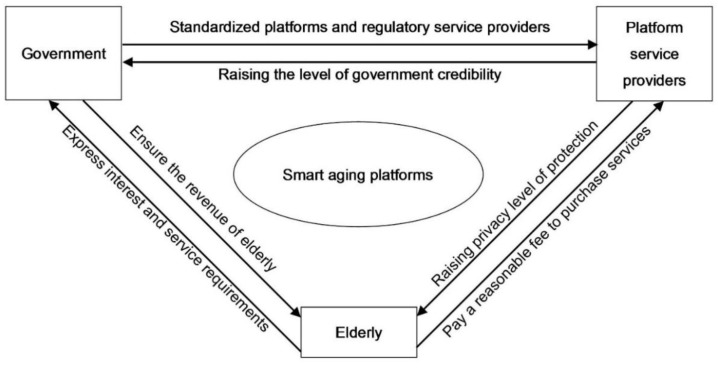
Subject relationship.

**Figure 2 ijerph-19-05778-f002:**
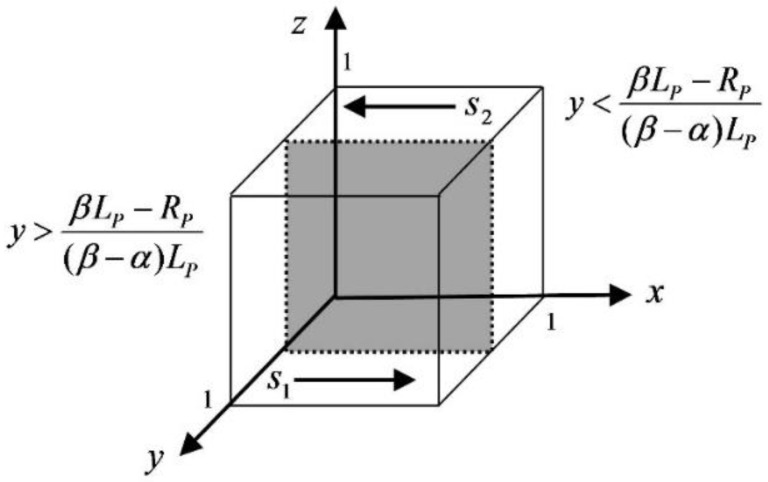
The elderly dynamics trend phase diagram.

**Figure 3 ijerph-19-05778-f003:**
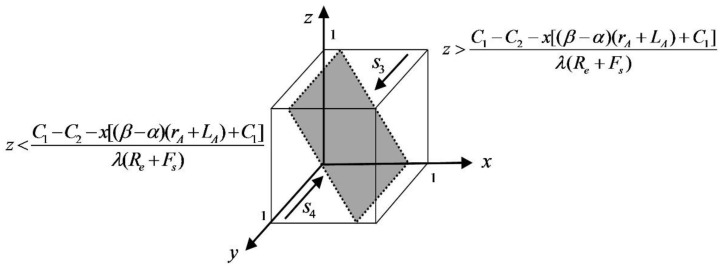
Phase diagram of dynamic trends in the platform service providers.

**Figure 4 ijerph-19-05778-f004:**
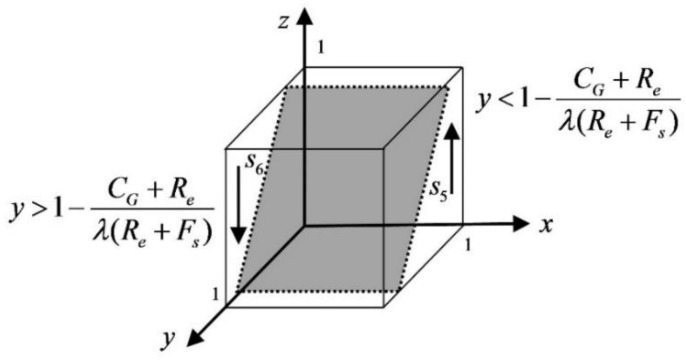
The government dynamic trend phase diagram.

**Figure 5 ijerph-19-05778-f005:**
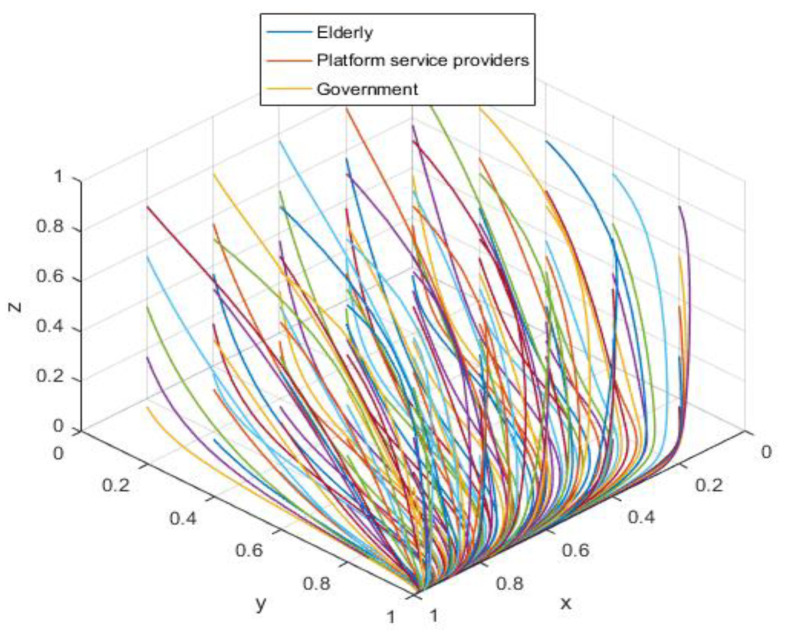
Ideal equilibrium state.

**Figure 6 ijerph-19-05778-f006:**
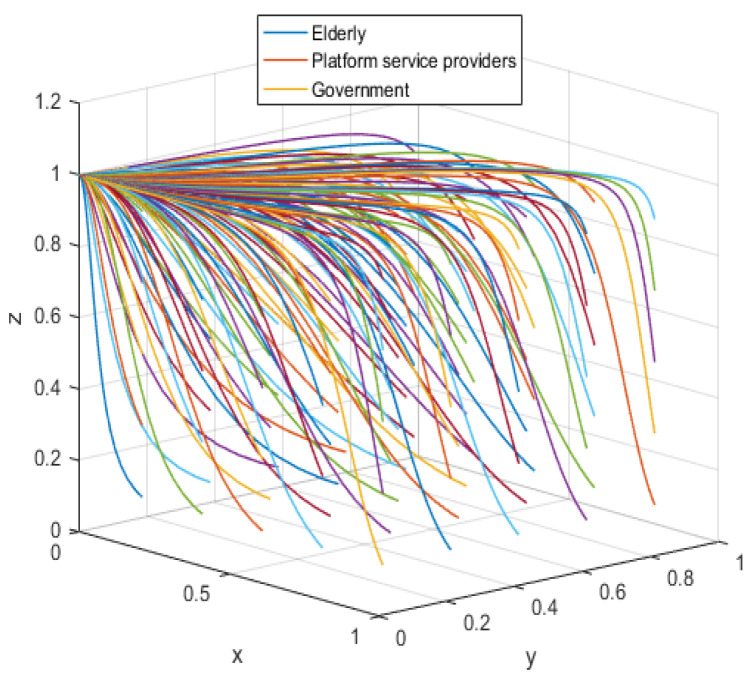
Initial equilibrium state.

**Figure 7 ijerph-19-05778-f007:**
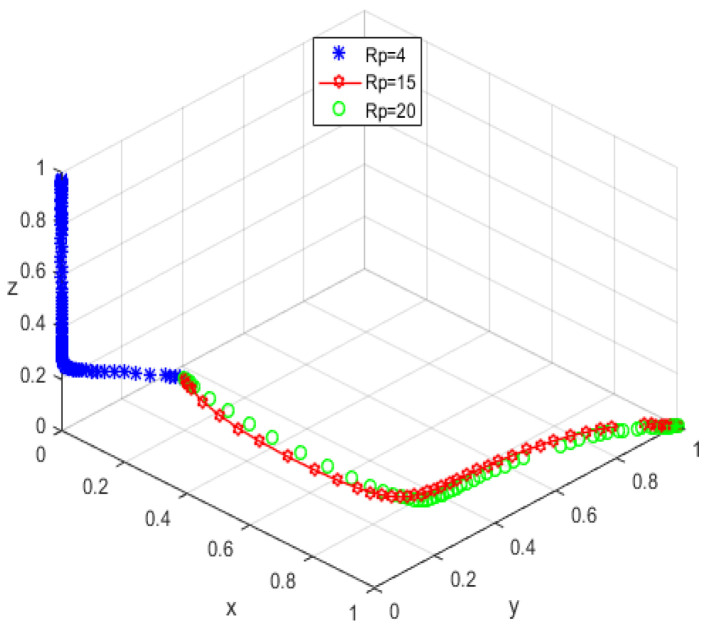
Trend graph for $R_{P}$.

**Figure 8 ijerph-19-05778-f008:**
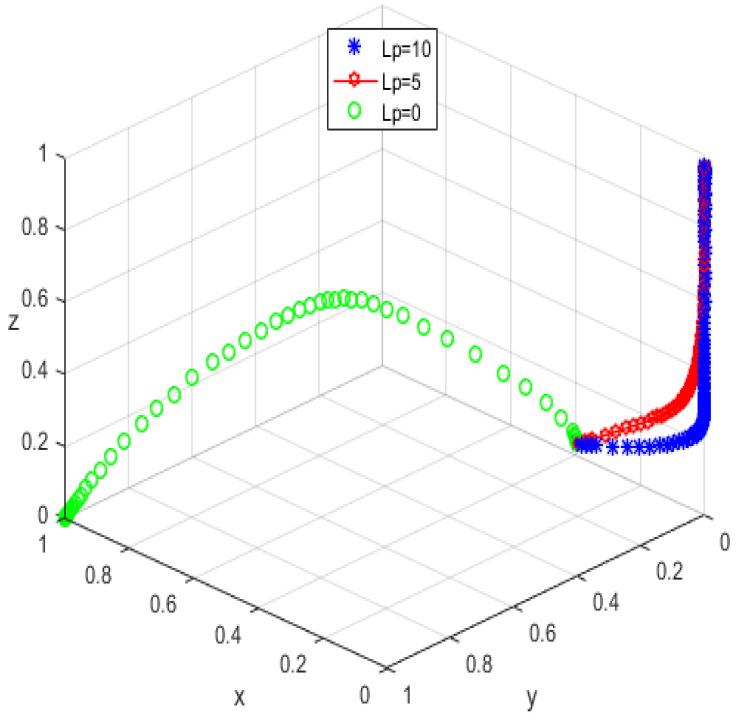
Trend graph for $L_{P}$.

**Figure 9 ijerph-19-05778-f009:**
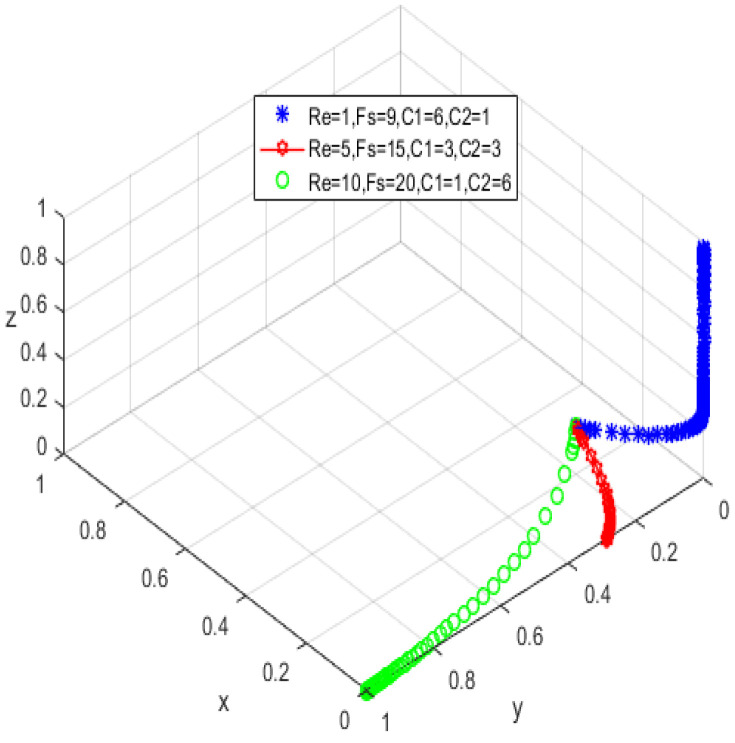
Trend graph for $R_{e}$, $F_{s}$, $C_{2}$, and $C_{1}$.

**Figure 10 ijerph-19-05778-f010:**
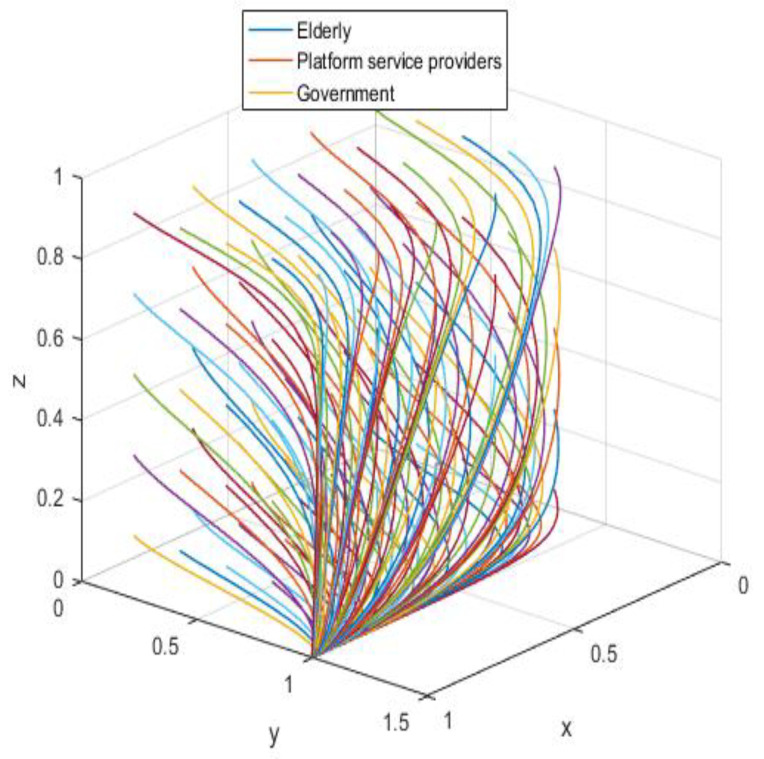
Optimized equilibrium state.

**Figure 11 ijerph-19-05778-f011:**
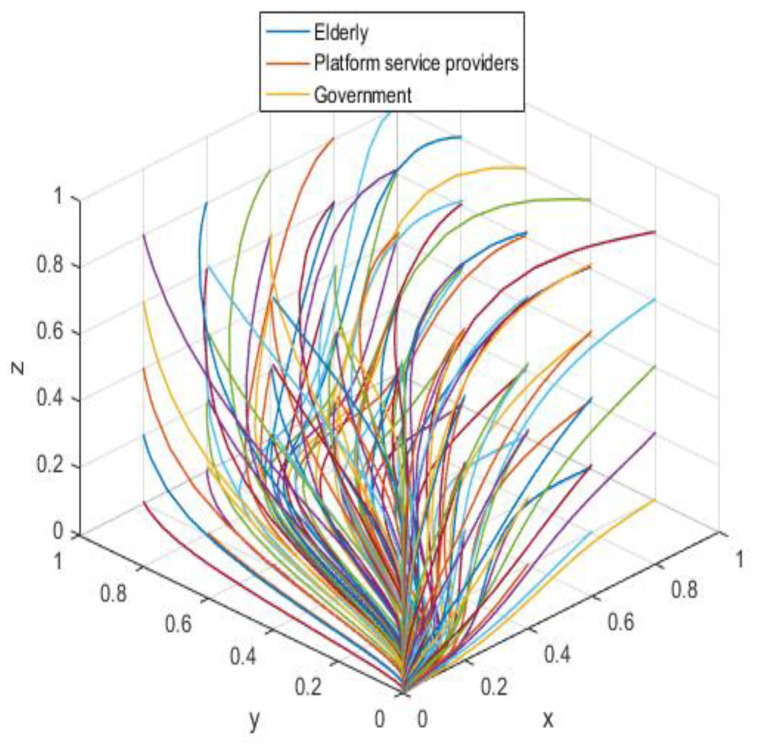
Initial equilibrium state.

**Figure 12 ijerph-19-05778-f012:**
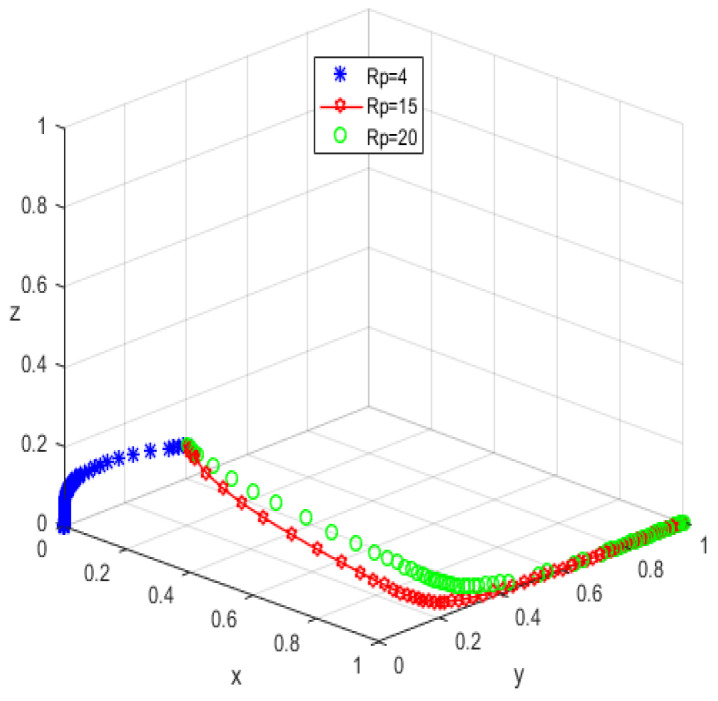
Trend graph for $R_{P}$.

**Figure 13 ijerph-19-05778-f013:**
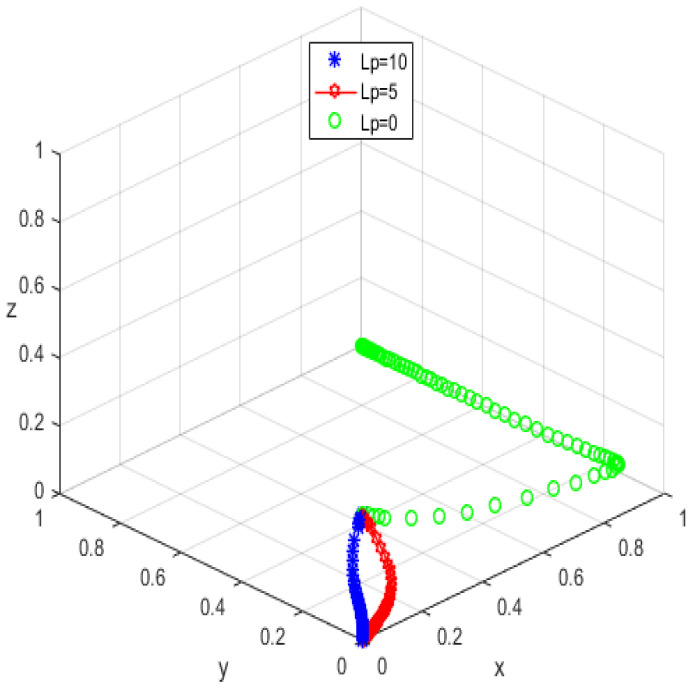
Trend graph for $L_{P}$.

**Figure 14 ijerph-19-05778-f014:**
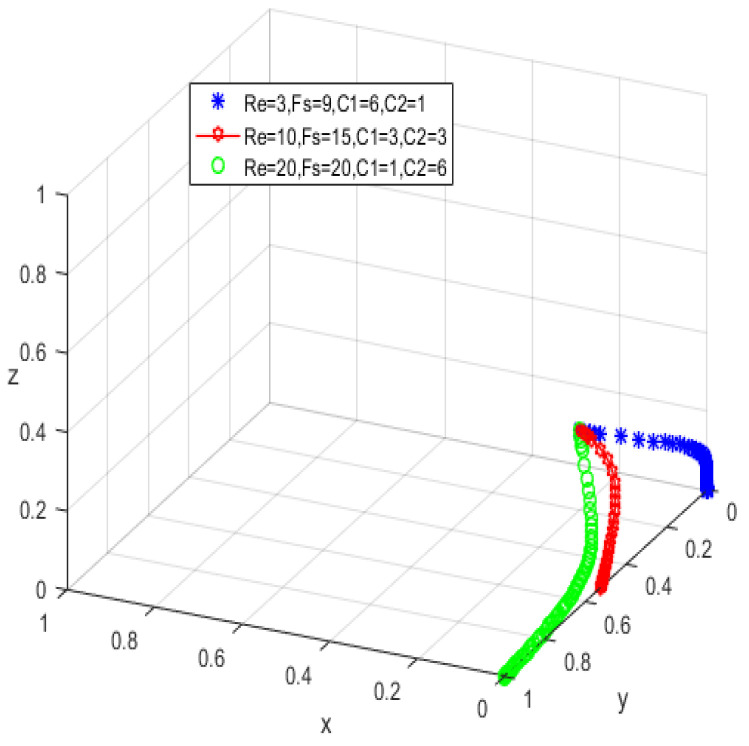
Trend graph for $R_{e}$, $F_{s}$, $C_{2}$, and $C_{1}$.

**Figure 15 ijerph-19-05778-f015:**
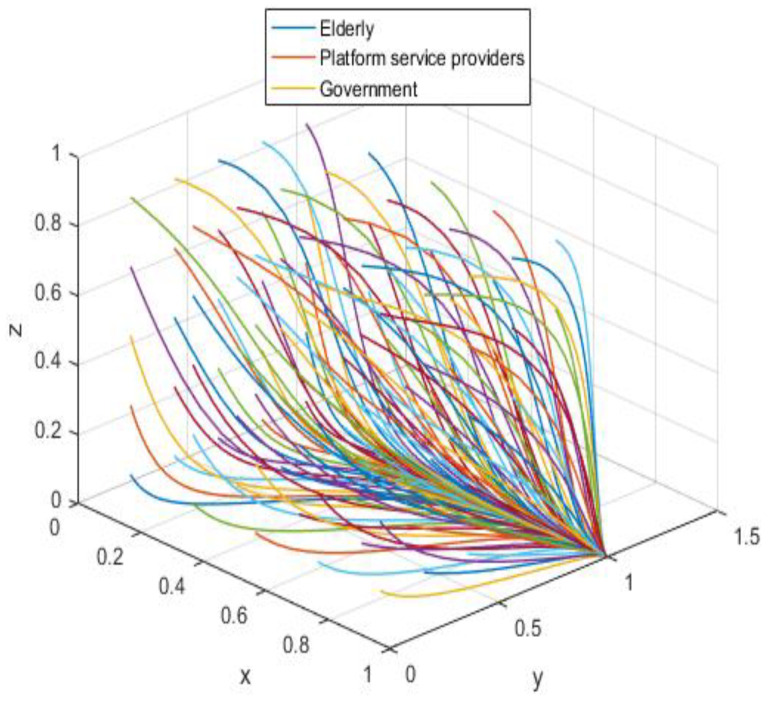
Optimized equilibrium state.

**Table 1 ijerph-19-05778-t001:** Main body parameters.

Main Body	Parameters	Explanatory Notes
**Elderly**	$R_{P}$	The elderly choose to participate in the supervision of service income, $R_{P}>0$.
	$L_{P}$	The elderly choose to participate in the supervision platform, and the loss caused by privacy leakage, $L_{P}>0$.
	$R_{t}$	The trust benefits brought by the government to the elderly during the high investment supervision, $R_{t}>0$.
	α	The probability of loss of privacy for platform service providers choosing high-quality protection, 0≤α≤1.
	β	The probability of loss of privacy for platform service providers choosing low-quality protection, $0\leq \dot{\alpha }\leq \beta \leq 1$.
**Platform service providers**	$R_{A}$	The fixed income brought by elderly people’s choice of participation in supervision, $R_{A}>0$.
	$C_{1}$	The cost of investment when choosing high-quality protection of privacy for platform service providers, $C_{1}>0$.
	$C_{2}$	The cost of input when platform service providers choose low-quality protection of privacy, $C_{2}>0$.
	$R_{e}$	Government subsidies for high investment supervision by the government and high-quality protection by service providers, $R_{e}>0$.
	$r_{A}$	Future benefits brought by platform service providers with the choice of high-quality protection privacy, $r_{A}>0$.
	$F_{s}$	Government fines for high investment supervision by the government and low-quality protection by service providers, $F_{s}>0$.
	$L_{A}$	The elderly choose to participate in the supervision platform, and the loss caused by privacy leakage, $L_{A}>0$.
**Government**	λ	The privacy probability of high investment supervision by that government and low-quality protection by service provider, 0≤λ≤1.
	T	Disclosure of privacy causes the government’s credibility to decline, T>0.
	$R_{G}$	The social benefits that elderly people choose to participate in the supervision platform and the service providers protect their privacy with high quality, $R_{G}>0$.
	$C_{G}$	Choosing the supervision cost of high investment supervision by the government, $C_{G}>0$.
	$L_{G}$	Reputation loss caused by elderly people’s choice of participation in the supervision platform and low-quality protection of privacy by service providers, $L_{G}>0$.

**Table 2 ijerph-19-05778-t002:** Matrix of benefits for game subjects.

	Participation (x)		Non-Participation (1-x)	
	High-Quality Protection (y)	Low-Quality Protection (1-y)	High-Quality Protection (y)	Low-Quality Protection (1-y)
**high input regulation** (z)	$R_{P}+R_{t}-\alpha L_{P}$,	$R_{P}+R_{t}-\beta L_{P}$,	$R_{t}$,	$R_{t}$,
	$\begin{array}{l} R_{A}+(1-\alpha )R_{A}\\ +R_{e}-C_{1}-\alpha L_{A} \end{array}$,	$\begin{array}{l} R_{A}+(1-\beta )r_{A}+(1-\lambda )R_{e}\\ -C_{2}-\beta L_{A}-\lambda F_{s} \end{array}$,	$R_{e}-C_{1}$,	$(1-\lambda )R_{e}-C_{2}-\lambda F_{s}$,
	$R_{G}-C_{G}-\alpha T-R_{e}$	$\lambda F_{s}-C_{G}-L_{G}-\beta T-(1-\lambda )R_{e}$	$-C_{G}-R_{e}$	$\lambda F_{s}-C_{G}-(1-\lambda )R_{e}$
**low input regulation** (1−z)	$R_{P}-\alpha L_{P}$,	$R_{P}-\beta L_{P}$,	0	0
	$R_{A}+(1-\alpha )R_{A}-C_{1}-\alpha L_{A}$,	$R_{A}+(1-\beta )r_{A}-C_{2}-\beta L_{A}$,	$-C_{1}$,	$-C_{2}$,
	$R_{G}-\alpha T$	$-L_{G}-\beta T$	0	0

**Table 3 ijerph-19-05778-t003:** Eigenvalues corresponding to system equilibrium points.

System Balance Point	Eigenvalue 1	Eigenvalue 2	Eigenvalue 3
$A_{1}(0,0,1)$	$R_{P}-\beta L_{P}$,	$\lambda (F_{s}+R_{e})+C_{2}-C_{1}$,	$\lambda (F_{s}+R_{e})-C_{G}-R_{e}$,
$A_{2}(0,1,1)$	$(\beta -\alpha )L_{P}+R_{P}-\beta L_{P}$,	$-[\lambda (F_{s}+R_{e})+C_{2}-C_{1}]$,	$C_{G}+R_{e}$,
$A_{3}(1,0,1)$	$-(R_{P}-\beta L_{P})$,	$(\beta -\alpha )(r_{A}+L_{A})+\lambda (R_{e}+F_{s})+C_{2}$,	$-[\lambda (F_{s}+R_{e})-C_{G}-R_{e}]$,
$A_{4}(1,1,1)$	$-[(\beta -\alpha )L_{P}+R_{P}-\beta L_{P}]$,	$-[(\beta -\alpha )(r_{A}+L_{A})+\lambda (R_{e}+F_{s})+C_{2}]$,	$C_{G}+R_{e}$,
$A_{5}(0,0,0)$	$R_{P}-\beta L_{P}$,	$C_{2}-C_{1}$,	$\lambda (F_{s}+R_{e})-C_{G}-R_{e}$,
$A_{6}(0,1,0)$	$(\beta -\alpha )L_{P}+R_{P}-\beta L_{P}$,	$C_{2}-C_{1}$,	$-C_{G}-R_{e}$,
$A_{7}(1,0,0)$	$-(R_{P}-\beta L_{P})$,	$(\beta -\alpha )(r_{A}+L_{A})+C_{2}$,	$\lambda (F_{s}+R_{e})-C_{G}-R_{e}$,
$A_{8}(1,1,0)$	$-[(\beta -\alpha )L_{P}+R_{P}-\beta L_{P}]$	$-[(\beta -\alpha )(r_{A}+L_{A})+C_{2}]$,	$-C_{G}-R_{e}$

**Table 4 ijerph-19-05778-t004:** Parameter assignment for $A_{8}(1,1,0)$.

Parameters	β	α	$L_{P}$	$R_{P}$	$r_{A}$	$L_{A}$	$C_{1}$	$C_{2}$	$C_{G}$	$R_{e}$	$F_{s}$	λ
**Numerical values**	0.6	0.4	5	10	6	4	5	7	8	10	6	0.5

**Table 5 ijerph-19-05778-t005:** Parameter assignment for $A_{1}(0,0,1)$.

Parameters	β	α	$L_{P}$	$R_{P}$	$r_{A}$	$L_{A}$	$C_{1}$	$C_{2}$	$C_{G}$	$R_{e}$	$F_{s}$	λ
**Numerical values**	0.6	0.5	10	4	2	1	6	1	0.5	1	9	0.2

**Table 6 ijerph-19-05778-t006:** Parameter assignment for $A_{5}(0,0,0)$.

Parameters	β	α	$L_{P}$	$R_{P}$	$r_{A}$	$L_{A}$	$C_{1}$	$C_{2}$	$C_{G}$	$R_{e}$	$F_{s}$	λ
**Numerical values**	0.6	0.5	10	4	2	1	6	1	0.5	3	9	0.2

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
