# Peer review of "A Regulatory Game Analysis of Smart Aging Platforms Considering Privacy Protection"

_ijerph, 2022, doi:10.3390/ijerph19095778_

Round 1
Reviewer 1 Report
- Please highlight the novelty of the manuscript in the abstract section.
- Line 58 “According to many scholars' surveys […]”, please provide citations.
- Lines 82-86. It’s not clear what exactly this manuscript is proposing. Please clearly state the proposed novelties.
- Section 3.1 is a bit hard to read. It might be better to separate the ideas into paragraphs.
- Line 262?
- Hypothesis 3, Please add a paragraph to highlight the connections between the parameters x, y, and z.
- Figures 2, 3, and 4 are a bit blurry.
- Pages 8, 9, and 10: I recommend the authors to have a closer loom on the way the equations are written.
- Lines 370-373, same as above.
- Section 4. It’s not clear to me which part is state-of-the-art and which part is proposed by the authors. To me, the presented stability analysis looks like a chapter from a well-known book/course.
- The manuscript is missing a comparison with state-of-the-art.
- The conclusion section is too long. Please draw the conclusions regarding what the manuscript managed to solve. The rest of the text can be moved in a discussion section.
The manuscript is well written in general, and presents in detail all algorithms. I would suggest the authors to provide a clear differentiation between state-of-the-art and the proposed method. There are also a few typos that must be corrected throughout the manuscript.
Author Response
Response to Reviewer 1 Comments
Point 1: Please highlight the novelty of the manuscript in the abstract section.
Response 1:
Original content:Therefore, there is significant value in exploring the regulation of smart aging platforms from the perspective of privacy protection. Based on evolutionary game theory and stability analysis, the evolu tionary path and dynamic equilibrium of the strategic choices of the elderly, platform service pro viders, and the government are explored, and MATLAB is used to carry out simulation analysis and game system optimization.
Updated content:Therefore, in this paper, based on the evolutionary game and Lyapunov theory, we discuss the willingness of elderly people to participate in regulation, the privacy protection status of platform service providers and the degree of government regulation, as well as the key factors affecting the equilibrium of the three-party game system, and conduct simulation analysis and game system optimization using MATLAB.
Modification instructions: It is introduced to study the willingness of elderly people to participate in regulation, the status of privacy protection of platform service providers and the degree of government regulation under the perspective of privacy protection, and to optimize the system. The innovative points of the article are named to reflect the novelty of the article. (See lines 15-20)
Point 2: Line 58 “According to many scholars' surveys […]”, please provide citations.
Response 2:
Original content:According to many scholars' surveys, it can be seen that …
Updated content:Up to now, there are still some problems that need to be solved urgently among the various subjects in the process of supervision of the smart elderly care platform.
Modification instructions: Since what is presented below is a study by scholars on the problems of the elderly, platform service providers and the government in the regulation process of the current smart senior care platform, the original line 58 has been slightly modified and is a summary of the whole paragraph. (See lines 69-70)
Point 3: Lines 82-86. It’s not clear what exactly this manuscript is proposing. Please clearly state the proposed novelties.
Response 3:
Original content:this paper will introduce evolutionary game theory to study the regulation strategy of smart aging platforms.
Updated content:this paper will introduce evolutionary game theory from the perspective of privacy protec-tion to discuss the supervision strategy of smart aging platforms. In addition to consider-ing the privacy concerns and benefit perceptions of the elderly, the willingness of the elderly to participate in supervision, the level of privacy protection of platform service providers and the degree of government supervision also plays an important role in the re-search on the supervision of smart aging platforms.
Modification instructions: In order to solve the above problems of the three parties in the supervision process of the smart pension platform, the optimal evolution and stability strategy is discussed around the construction of an evolutionary game system by the three parties, and the relationship between the willingness of the elderly to participate in supervision, the level of privacy protection of platform service providers and the degree of government supervision is pointed out importance. (See lines 94-99)
Point 4: Section 3.1 is a bit hard to read. It might be better to separate the ideas into paragraphs.
Response 4:
Modification instructions: In section 3.1, the original whole paragraph is divided into two paragraphs, making the content more hierarchical.
Point 5: Line 262?
Response 5:
Modification instructions: Line 262 of the original text (i.e., now line 320) has been transposed causing the entire hypothesis 3 to be subdivided and has been consolidated into one paragraph.
Point 6: Hypothesis 3, Please add a paragraph to highlight the connections between the parameters x, y, and z.
Response 6:
Modification instructions: Hypothesis 3 is added to explain the relationship between x, y, and z. x, y, and z represent the decision probability level of the tripartite subject. The first paragraph of Section 3.1 describes the relationship between the tripartite subjects represented by x, y and z. (See lines 315-324)
Point 7: Figures 2, 3, and 4 are a bit blurry.
Response 7:
Modification instructions: The resolution of Fig. 1, Fig. 2, Fig. 3 and Fig. 4 was adjusted to 300 dpi to make the images look more clear.
Point 8: Pages 8, 9, and 10: I recommend the authors to have a closer loom on the way the equations are written.
Response 8:
Modification instructions: The formulas (1) to formula (15) were aligned by inserting a table, and the formula serial numbers were right-aligned in the table to solve the problem of irregular formulas.
Point 9: Lines 370-373, same as above.
Response 9:
Modification instructions: Adjusted the problem of misalignment of constraints and sentences in lines 370-373 (i.e., now lines 434-436) of the original text.
Point 10: Section 4. It’s not clear to me which part is state-of-the-art and which part is proposed by the authors. To me, the presented stability analysis looks like a chapter from a well-known book/course.
Response 10:
Modification instructions: Regarding the method research in Section 4, the unilateral subject stability analysis in Section 4.1 is a state-of-the-art analysis of evolutionary stability strategies. In addition to analyzing the willingness of the elderly to participate in supervision, it also considers platform services The level of commercial privacy protection and the degree of government supervision (that is, the second innovation point of this paper). In the stability analysis of the tripartite system in Section 4.2, based on the Lyapunov theory, this paper proposes to transform the unstable point to the stable point, which is the optimal ESS of the tripartite system (that is, the third innovation point of this article).
Point 11: The manuscript is missing a comparison with state-of-the-art.
Response 11:
Modification instructions: It is summarized here that the research methods of the above literature review on the regulation of the wisdom elderly platform mainly focus on the technical implementation and application analysis, and more on the elderly as the research subject, but the level of privacy protection of the platform service providers and the degree of government regulation research are not taken into account, nor is the long-term nature of privacy leakage, and there is information asymmetry as well as limited rationality among the three (already in hypothesis 2 stated), and therefore qualify for the evolutionary game approach. Thus, a comparison between the proposed method and the state-of-the-art method is reflected here. (See lines 238-247)
Point 12: The conclusion section is too long. Please draw the conclusions regarding what the manuscript managed to solve. The rest of the text can be moved in a discussion section.
Response 12:
Modification instructions: According to the first reviewer's opinion, the length of Section 6 was shortened, the original second paragraph was deleted, and the innovation was reflected in the last paragraph of Section 2.

Reviewer 2 Report
The manuscript titled “A regulatory game analysis of smart aging platforms considering privacy protection” proposed an investigation on the regulation of smart aging platforms from the perspective of privacy protection. In the framework of evolutionary game theory and stability analysis, the inter-relationships between the three actors involved in the creation and fruition of smart aging platforms (i.e., elderly, platform service providers, and government) were explored, and the evolutionary path and dynamic equilibrium of the strategic choices of each actor were evaluated. MATLAB was used to perform simulation analysis for scenarios with different parameters, and several models were performed and compared. Results showed that the parameters ensuring the optimal evolutionary stability strategy were participation, high-quality protection, and low-investment supervision. The key factors affecting the game system with three actors were the service income of the elderly, the loss of privacy leakage, the investment cost of service providers, and the amount of government rewards and punishments. Authors discussed their results in light of previous literature as well as practical implications of their findings.
I carefully read the manuscript, and I think it may be of interest for the readers of IJERPH. The manuscript is very well-written and properly addressed the interesting issue of the factors which can facilitate the spreading and the use of smart aging platforms. It is also relevant that Authors used a game theory approach to analyze the phenomenon and to identify the key factors as well as the optimal strategy between the three main actors of the system.
It was a pleasure for me to read the manuscript. The introduction section as well as the aims of the study are clear and detailed. The methodology and the simulation analysis are very rigorous and accurate, as well as the explanations provided in the discussion section. I also appreciated the practical implications for each of three actors involved in the model.
I have no further considerations or remarks, and I think that the present manuscript could be published as it is.
Author Response
Response to Reviewer 2 Comments
Response :
Dear reviewer,
Thank you very much for reviewing my paper, and also thank you for your evaluation and appreciation of this article, which also adds more confidence and motivation to me in the future research path.
Although you did not suggest relevant revisions to me, I made the revisions in conjunction with the comments of the other two reviewers. The main modifications are summarized as follows:
- Relevant grammar checks and formula corrections were made to the full text.
- The abstract has been optimized.
- The contribution points of this paper are added at the end of the literature review, which is also the innovation of this paper.
- State the research question, and indicate the motivation and goals of the research, as well as the novelty of the method.
- Parameter analysis and description are carried out for Figure 2, Figure 3, and Figure 4.
- Figures 1 to 4 are changed to 300dpi, which improves the clarity of the pictures.
Yours,
Tengfei Shi

Reviewer 3 Report
The manuscript addresses an important issue of game analysis. However, some problems still exist, and the manuscript needs to be improved by considering the following comments:
It is recommended to use a professional native English-speaking editor. Papers with less than excellent English will not be published even if technically perfect. -
- The Abstract is clear and comprehensive but Needs to highlight the result numerically (brief) in the abstract.
- The problems of this work are not clearly stated. There is ambiguity in statement understanding.
- What is the motivation for the proposed work? Research gaps and objectives of the proposed work should be clearly justified.
- Describe the major contributions of the study.
- As privacy protection is issue in the era of big data, for this you should describe the importance of big data for analysis with the latest references, see for example this paper which would help you
- https://doi.org/10.3390/su131910743
- Explain what kind of simulations will be provided in Section 4 at the beginning of the section
- In the results section, some figures and tables need more description.
Author Response
Response to Reviewer 3 Comments
Point 1: It is recommended to use a professional native English-speaking editor.
Response 1:
Modification instructions: Thanks to the advice of the reviewer, the full text has been carefully checked and the relevant incorrect sentences and words have been fixed.
Point 2: The abstract is clear and comprehensive but needs to highlight the result numerically (brief) in the abstract.
Response 2:
Modification instructions: Removed "the optimal evolutionary stability strategy" and added a digital description of the results. (See lines 20-21)
Point 3: The problems of this work are not clearly stated. There is ambiguity in statement understanding.
Response 3:
Modification instructions: Regarding the problem statement of the article, the second paragraph of the introduction describes the problems that exist in the regulation of the current smart senior care platform and analyzes the conflict of interest that exists between the elderly, the platform service provider, and the government (see lines 69-84). In addition, the second section concludes with a summary of previous research on the regulation and privacy of smart senior care platforms and identifies the problems that exist (see lines 238-246).
Point 4: What is the motivation for the proposed work? Research gaps and objectives of the proposed work should be clearly justified.
Response 4:
Modification instructions: The motivation and objectives of this paper are mentioned at the beginning of the introduction. With the aging of our society, the problem of imbalance between supply and demand of the elderly needs to be solved, and the smart elderly platform can alleviate this contradiction, but the privacy protection is still not good enough, which will cause some privacy concerns to the elderly. Therefore, this paper needs to solve the supervision problem of smart senior care platform from the perspective of privacy protection, aiming to reduce the privacy concern of the elderly and improve the supervision of the smart senior care platform, which is the motive of this paper. (see lines 66-68)
Point 5: Describe the major contributions of the study.
Response 5:
Modification instructions: The main contributions made in this paper are summarized, as well as the three innovations involved in this paper.( see lines 253-263)
Point 6: As privacy protection is issue in the era of big data, for this you should describe the importance of big data for analysis with the latest references, see for example this paper which would help you.
https://doi.org/10.3390/su131910743
Response 6:
Modification instructions: After referring to your advice, I learned a lot from the article you gave me and quoted this article you provided. The 18th reference was replaced with the literature provided in the expert's advice, reflecting the current research status of privacy data protection. ( see lines 190-192)
Point 7: Explain what kind of simulations will be provided in Section 4 at the beginning of the section
Response 7:
Modification instructions: The fourth section analyzes stability, determines the evolutionary stability strategy of uni-lateral and tripartite systems, and points out the key factors that affect the deci-sion-making behavior of tripartite entities. In the fifth section, simulation analysis and system optimization are carried out by MATLAB. The specific content is the initial state simulation, parameter change simulation and optimal state simulation. ( see lines 137-141)
Point 8: In the results section, some figures and tables need more description.
Response 8:
Modification instructions: In the unilateral subject stability analysis of the elderly, platform service providers, and the government, a parametric analysis was added in combination with graphics to more vividly depict Figure 2, Figure 3, and Figure 4. ( see lines 352-358, 375-383 and 398-404)
